# Scalable Multimer Structure Prediction using Diffusion Models

**Peter Pao-Huang**
Department of Computer Science
University of Illinois Urbana-Champaign
`ytp2@illinois.edu`

**Bowen Jing**
CSAIL
Massachusetts Institute of Technology
`bjing@mit.edu`

**Bonnie Berger**
CSAIL & Department of Mathematics
Massachusetts Institute of Technology
`bab@mit.edu`

## Abstract

Accurate protein complex structure modeling is a necessary step in understanding the behavior of biological pathways and cellular systems. While some works have attempted to address this challenge, there is still a need for scaling existing methods to larger protein complexes. To address this need, we propose a novel diffusion generative model (DGM) that predicts large multimeric protein structures by learning to rigidly dock its chains together. Additionally, we construct a new dataset specifically for large protein complexes used to train and evaluate our DGM. We substantially improve prediction runtime and completion rates while maintaining competitive accuracy with current methods.

## 1 Introduction

The structure of proteins determines their function. In recent years, the problem of predicting protein structure has seen significant improvement with models like AlphaFold2 [13]. While accurate modeling of proteins is important, proteins rarely operate in isolation. They commonly require assembling into multimeric complexes to gain function, either in a collection of identically repeating protein chains (homomers) or a collection of differing protein chains (heteromers). Therefore, the structure of a protein chain is often insufficient in understanding its behavior without knowledge of its larger multimeric structure.

Taking inspiration from the recent success of *diffusion generative models* in structural biology problems [8, 21, 22], we frame protein complex structure prediction as a generative modeling task. Given the structure of protein chains (predicted by ESMFold or other protein structure prediction models), we propose a score-based diffusion model that learns to rigidly dock the chain structures to form the larger complex. We also develop a new dataset called LPC-Dataset for **l**arge **p**rotein **c**omplex structures (with complex sizes from 5 to 30 chains).

We train on the LPC-Dataset and evaluate our method on the test set provided by Bryant et al. [2] (with complex sizes from 10 to 30 chains). Empirically, we achieve comparable prediction quality while outperforming previous work in inference runtime and completion rates (defined as being able to predict the structure of the *entire* complex).

## 2 Background

**Multimeric Protein Structure Prediction.** Current work on modeling the structure of protein complexes can be divided into two approaches: (1) traditional docking [1, 6, 10, 18, 4] and (2) multimeric structure prediction using deep learning like AlphaFold2-Multimer [5]. More recent deep learning approaches address rigid-body docking [14, 7] but have not scaled beyond dimers. AlphaFold2-Multimer still largely outperforms these works while covering complexes with up to 9 chains [5].

However, the critical issue that still remains in current work is that they can only operate on smaller complexes (within the range of 2-9 chains), leaving a need for a model that can scale to larger complexes.

**Diffusion Generative Models.** Ho et al. [9] have demonstrated the impressive capabilities of diffusion generative models in creating realistic images. Excitingly, a recent wave of methods have adopted this class of generative models outside of images to structural biology problems, many achieving competitive results in tasks like protein folding [12] and design [23], protein-ligand docking [3], and others [11, 15, 19].

Figure 1: Inference procedure involves *multi-body* rigid docking to generate protein complex conformation.

These generative models operate by iteratively adding noise to destruct data to a simple prior distribution (forward diffusion process) and then learning to restore samples drawn from the prior to the initial data distribution (reverse diffusion process). Song et al. [20] formulates the forward and reverse diffusion processes in the continuous case as stochastic differential equations in the form of $dx = f(x,t)dt + g(t)dw$ and $dx = [f(x,t) - g(t)^2 \nabla_x \log p_t(x)]dt + g(t)dw$, respectively. To carry out the reverse diffusion process, we learn to approximate the $\nabla_x \log p_t(x)$ term (aka the *score*) at all times $t \in [0, \infty]$, typically using a neural network that we denote as $\mathbf{s}_{\boldsymbol{\theta}}$.

## 3 Method

We aim to obtain the structure of a large protein complex given each chain's sequence. We first assume we have the structure of each chain as input. These structures can be given by the ground-truth crystal structure or predicted by a protein structure prediction model. Our method then learns to rigidly and simultaneously dock the individual protein structures together using a diffusion generative model. We consider the resulting conformation generated by our docking procedure as the final predicted structure of the protein complex.

### 3.1 Protein Complex Representation

We conduct a coarse-graining procedure that groups every $k$ residue together based on sequence proximity. For a given complex $s$ chains, we denote the non-coarse grained complex as $\mathcal{G} = \{\mathcal{V}_0, \ldots, \mathcal{V}_s\}$ and node features $\mathcal{V}_i = \{\mathbf{v}_1, \ldots, \mathbf{v}_{n_i}\}$ where $\mathbf{v}_i \in \mathbb{R}^{(m+3)}$ (three is the 3D coordinates, $m$ is the embedding dimension, and $n_i$ is the number of residues in chain $i$). We typically set $m$ as the dimension of the residue embedding from a protein language model like ESM-2 [16] or OmegaPLM [24]. We then construct our coarse-grained complex graph, $\hat{\mathcal{G}}$, where each coarse-grained node is the average position and embeddings of the residues grouped to that node:

$$\hat{\mathcal{G}} = (\{\hat{\mathcal{V}}_1, \ldots, \hat{\mathcal{V}}_s\}, \mathcal{E}), \quad \hat{\mathcal{V}}_i = \{\hat{\mathbf{v}}_1, \ldots, \hat{\mathbf{v}}_{\lceil n_i/k \rceil}\}, \quad \hat{\mathbf{v}}_{j \in 1 \ldots \lceil n_i/k \rceil} = \frac{\sum_{l=jk}^{j(k+1)} \mathbf{v}_j}{k} \quad (1)$$

For clarity in the proceeding sections, we will refer to $\hat{\mathcal{V}}_{x,i} \in \mathbb{R}^3$ as the position of chain $i$ and $\hat{\mathcal{G}}_x \in \mathbb{R}^{s \times 3}$ as the positions of all chains $\hat{\mathcal{V}}_i$ in the complex $\hat{\mathcal{G}}$. Because of our coarse-graining procedure, we can feasibly construct a complete graph, which is important in diffusion generative

models—at high noise levels of the diffusion process, nodes need to communicate with physically distant nodes to learn accurate scores.

## 3.2 Diffusion Over $SE(3)$

Since we consider protein complex structure prediction in the *many-body* rigid-docking setting, each chain $\hat{\mathcal{V}}_i$ in a complex $\hat{\mathcal{G}}$ is constrained to group elements in $SE(3)$—the group representing rigid-body motions. Therefore, we aim to learn a density over $SE(3)^s$ where $s$ is the number of chains.

Following the framework established by Yim et al. [25] for diffusion generative models on $SE(3)$, we define the following forward variance-exploding SDEs separately for translation and rotation:

$$d\hat{\mathcal{G}}_{\mathbf{x}} = \sqrt{d\sigma_{tr}^2(t)/dt}d\mathbf{w}, \quad d\hat{\mathcal{G}}_{\mathbf{x}} = \sqrt{d\sigma_{rot}^2(t)/dt}d\mathbf{w} \tag{2}$$

where $\mathbf{w}$ is the Brownian motion over $T(3)^s$ and $SO(3)^s$, respectively. The resulting translation diffusion kernel is $p_{tr}(\hat{\mathcal{G}}_{x,t}|\hat{\mathcal{G}}_{x,0}) = \mathcal{N}(\hat{\mathcal{G}}_{x,t}; \hat{\mathcal{G}}_{x,0}, \sigma_{tr}(t))$ whose score is defined as $\nabla \log p_{tr}(\hat{\mathcal{G}}_{x,t}|\hat{\mathcal{G}}_{x,0}) = \frac{-(\hat{\mathcal{G}}_{x,t}-\hat{\mathcal{G}}_{x,0})}{\sigma_{tr}^2(t)}$. We refer readers to Corso et al. [3] for the rotation kernel and score calculations over $SO(3)$.

## 3.3 Score-Matching Degeneracy in Complexes w/ Multiple Identical Chains

Homomers and proteins with multiple identical chains commonly appear in nature. These types of protein complexes pose a unique challenge for denoising score-matching as illustrated in 2. We introduce two techniques that transform scores to resolve score-matching degeneracy. Recall that the score-matching objective, in practice, follows

$$L_\theta(\hat{\mathcal{G}}_t, \hat{\mathcal{G}}_0) = \left\| \mathbf{s}_{\boldsymbol{\theta}}(\hat{\mathcal{G}}_t) - \mathbf{s}_{gt}(\hat{\mathcal{G}}_{x,t}, \hat{\mathcal{G}}_{x,0}) \right\|_2^2, \quad \mathbf{s}_{gt}(\hat{\mathcal{G}}_{x,t}, \hat{\mathcal{G}}_{x,0}) = \nabla \log p_{tr}(\hat{\mathcal{G}}_{x,t}|\hat{\mathcal{G}}_{x,0}) \tag{3}$$

Note that the true score $\mathbf{s}_{gt}$ is defined only using the positions while the *score model* $\mathbf{s}_{\boldsymbol{\theta}}$ takes as input both the positions and the embeddings. We aim to revise the computation of $\mathbf{s}_{gt}$ to eliminate score-matching degeneracy.

**Hungarian Score Assignments.** We can view the degeneracy issue as a bipartite matching problem where we want to match each noised chain with the optimal ground truth chain in the same set of identical chains. Under this view, for any given noised state of the protein complex, there is a single possible set of scores. We first define our cost matrix as the negative log-likelihood, $-\log(p(\hat{\mathcal{V}}_{x,i,t} \mid \hat{\mathcal{V}}_{x,j,0}))$, between all pairs of noised and ground truth states, $i$ and $j$, of the same identity. We then run the Hungarian algorithm to find the optimal bipartite matching that minimizes the cost matrix and use that assignment to compute new scores. However, an issue with this technique is that the resulting score vector field is non-differentiable, which prompts us to propose another alternative procedure.

**Weighted-Sum Score.** Instead of using ground truth scores, $\nabla \log p_{0t,i}(\hat{\mathcal{V}}_{x,i,t} \mid \hat{\mathcal{V}}_{x,i,0})$, we recalculate the score of chain $i$ as the weighted sum of the potential scores to all other identical chains (we denote the set of identical chains corresponding to a $\hat{\mathcal{V}}_{x,i,t}$ as $g$).

$$\mathbf{s}_{gt}(\hat{\mathcal{V}}_{x,i,t}) = \sum_{\hat{\mathcal{V}}_{x,j,t} \in g} \frac{\nabla \log p(\hat{\mathcal{V}}_{x,j,t} \mid \hat{\mathcal{V}}_{x,j,0}) p(\hat{\mathcal{V}}_{x,j,t} \mid \hat{\mathcal{V}}_{x,j,0})}{\sum_{\hat{\mathcal{V}}_{x,k,t} \in g} p(\hat{\mathcal{V}}_{x,k,t} \mid \hat{\mathcal{V}}_{x,k,0})} \tag{4}$$

## 3.4 Training & Inference Procedure

Given the ability to sample and compute scores for the $SO(3)^s$ and $T(3)^s$ diffusion kernels, we train our base score model using the following loss:

$$y_r^{(i)} = \nabla \log p_{tr}(\Delta\hat{\mathcal{V}}_{x,t}|\hat{\mathcal{V}}_{x,0})), y_R^{(i)} = \nabla \log p_{rot}(\Delta\mathbf{R}_t|\mathbf{R}_0)) \tag{5}$$

$$L_\theta = \frac{\sum_{i=0}^s ||\alpha_i - y_r^{(i)}||^2 + ||\beta_i - y_R^{(i)}||^2}{s} \tag{6}$$

where $\alpha_i$ is the predicted translation score, $\beta_i$ is the predicted rotation score for chain $i$, and $\mathbf{R}$ is the rotation. Note that the Hungarian and weighted-sum scores instead define $y^{(i)}$ as the computation of $s_{gt}$. We use ground truth chain structures for training and switch to ESMFold predicted chain structures for inference.

With a trained score model $s_\theta$, we conduct our inference procedure via 1 in the Appendix.

## 3.5 Architecture

We use a $SE(3)$-equivariant convolutional network for our score model, $s_\theta$, that is adopted from Corso et al. [3]. The score model, $s_\theta(\hat{\mathcal{G}}, t)$, takes as input the coarse-grained protein complex graph and time. Since our protein complex graph is complete with respect to both the nodes within a chain and between chains, we want to disambiguate by defining two types of edges: intra-edges (edges in the same chain) and inter-edges (edges between different chains). We then have intermediate convolutional layers that alternate between convolving between intra-edges and inter-edges.

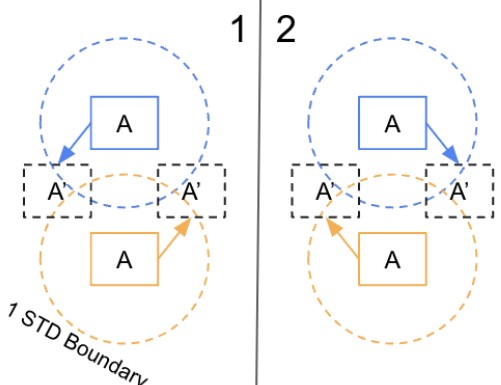

In the final layer, we convolve the node features in each chain around the chain's center of mass. For each chain, we output a translation and rotation score vector in the tangent space of the associated manifold. These scores are by construction $SE(3)$-equivariant with respect to the input graph.

Figure 2: Example of score matching degeneracy during the diffusion process for a protein complex. Given the ground truth state of two identical chains $A$, the diffusion process places them at noised states $A'$. During the reverse diffusion process, the score model can predict two sets of scores (left and right, signified by the arrows $A' \rightarrow A$), both of which are valid. This degeneracy can result in high variance for the score estimator.

## 4 Experiments

We curate a custom dataset (LPC-Dataset) for **l**arge **p**rotein **c**omplex structures by filtering through all PDB biological assemblies. We then train and evaluate our model and two variants using the LPC-Dataset. The base model and its variants are trained for 60 epochs.

### 4.1 LPC-Dataset Collection Procedure

We begin with collecting our training and validation data. Using the PDB biological assemblies archive as a starting dataset, we first remove all protein complexes with non-polypeptide subunits. We then remove any chains with less than 20 residues but keep the rest of the complex. Additionally, any complexes with a number of chains outside of the range $[5, 30]$ are removed.

A test set with 175 complexes is obtained from MoLPC [2] where all complexes have between $[10, 30]$ chains. Due to our aforementioned filtering procedure, 5 complexes were removed from the test set resulting in a final test set of 170 complexes. To ensure that our training and validation set does not share any similar complexes with the test set, we use MMSeq2 to cluster the chains of all complexes in the training/validation and test sets by $40\%$ identity (common threshold [17] for homologous structures). We denote complex $A$ as a subset of complex $B$ if all chains of $A$ have a similar chain in $B$. We then remove any complex in the training and validation set that is a subset of another complex in the test set. We divide the remaining dataset into the respective train and validation splits by randomly choosing $5\%$ to be our validation set and the remaining complexes to be our train set. LPC-Dataset constitutes the collective training, validation, and test set.

### 4.2 Protein Complex Structure Prediction

Our model is evaluated on the test set from LPC-Dataset. Note that since other methods can only assemble a portion of the entire test set, we penalize the complexes that the methods cannot assemble

Table 1: Median/Mean performance metrics evaluated on the test set. We denote the Hungarian variant as "V1" and the weighted-sum variant as "V2." The asterisk signifies protein model predicted chain structures as input while non-asterisks use ground-truth chain structures. [†]Note that the RMSD reported for MoLPC is only evaluated on the subset of the test set it can completely assemble.

| METRICS | TM-SCORE ↑ | RMSD ↓ | % COMPLETE ↑ |
|---|---|---|---|
| BASE* | 0.19/0.20 | 9.68/9.91 | 100.0 |
| BASE | 0.28/0.32 | 9.92/10.05 | 100.0 |
| V1 | 0.18/0.20 | 9.37/9.43 | 100.0 |
| V2 | 0.23/0.25 | 9.93/10.11 | 100.0 |
| MoLPC* | 0.20/0.39 | 7.12/7.18[†] | 51.8 |
| HD | 0.17/0.23 | — | 45.3 |

to a default TM-score of 0.17 (max value associated with a random prediction [26]). We also use the metrics reported in Bryant et al. [2] for both Haddock (HD) and MoLPC. The results are detailed in 1. Additionally, we include the performance of the two variants (one trained with the Hungarian scores and the other trained with the weighted-sum scores).

**Assembly Completion**. It is important that we can obtain the structure of the entire complex successfully. However, for both MoLPC and HD, they were only able to fully assemble around half of the complexes in the test set either because of superposition clashes or OOM issues. Due to the nature of our method, we are able to completely assemble all of the complexes included in the test set.

**Quality of Prediction**. As shown in 1, our method is comparable with the results of HD and MoLPC. We believe that the Base Model* can be improved by training on ESMFold predicted structures that are RMSD-aligned with the ground truth position and orientation rather than training on ground truth structures. We also see that the proposed variants do not clearly outperform the base model, which could imply that the score-matching degeneracy noted earlier is not a significant issue in practice.

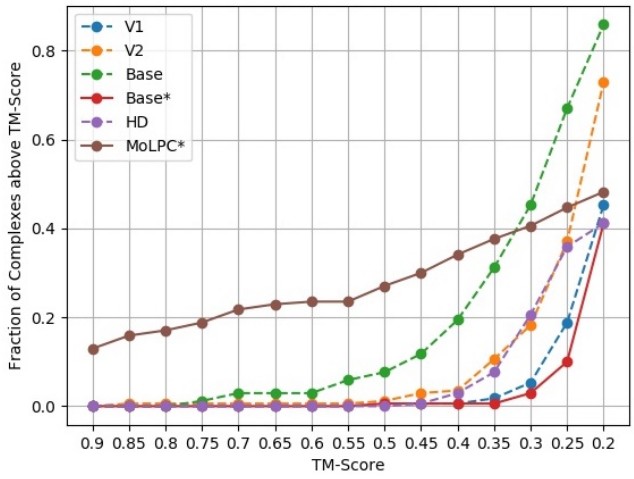

Figure 3: Cumulative density of predicted complexes from each method with respect to thresholds of TM-score. Dotted lines denote methods that use ground-truth chain structures.

**Runtime**. To predict the individual chains of the complex, we use ESMFold, which roughly takes a few minutes to predict all chains of a complex. We refer the reader to the ESMFold paper for precise runtime benchmarking. These chains are then used as input to our base model, which takes on average 22.67 seconds for inference per complex across the full test set. Collectively, the end-to-end pipeline for predicting the structure of a large complex is substantially faster than MoLPC ($39 - 52$ hours).

## 5   Conclusion

Modeling the structure of protein complexes is imperative to better understand the intricacies of our biology. While not yet state-of-the-art, our method runs significantly faster and can more reliably predict the entire complex than previous work. Additionally, we identify and attempt to address key issues with adapting diffusion generative models to large protein complex structure prediction. There are many fruitful directions for future work that could improve the results of the proposed method including (1) using a force-field relaxation to obtain more accurate protein interfaces and local topology and (2) docking dimers/trimers rather than individual chains.

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

# A Appendix

We highlight the inference procedure of our diffusion model and the distribution of complex sizes in our training & validation dataset.

---

**Algorithm 1** Multi-Body Docking Inference Procedure

---

    **Input:** Protein chain structures $\{\hat{\mathcal{V}}_i\}$ of size $s$, steps $N$, score model $s_\theta$
    **Output:** predicted protein complex conformation $\{x_0\}$
    Initialize schedule $\phi_{tr} = [\sigma_{tr}^2(\frac{1}{N}), \ldots, \sigma_{tr}^2(\frac{N}{N})]$
    Initialize schedule $\phi_{rot} = [\sigma_{rot}^2(\frac{1}{N}), \ldots, \sigma_{rot}^2(\frac{N}{N})]$
    **for** $i = 0$ **to** $s$ **do**
        Center $x_i$ at $(0, 0, 0)$
        $\Delta r_1 \sim p_{tr}(\Delta r_1), \Delta R_1 \sim p_{rot}(\Delta R_1)$
        Update $x_i$ with translation $\Delta r_1$ and rotation $\Delta R_1$
    **end for**
    **for** $i = N$ **to** $1$ **do**
        $t = \frac{i}{N}, \Delta t = \frac{i}{N} - \frac{i-1}{N}$
        $\alpha, \beta = s_\theta(\{x_i\}, t)$ where $\alpha, \beta \in \mathbb{R}^{3s}$
        $\mathbf{z_{tr}} = \mathcal{N}(0, \phi_{tr,i}), \mathbf{z_{rot}} = \mathcal{N}(0, \phi_{rot,i})$
        $\Delta r = \alpha(\phi_{tr,i} - \phi_{tr,i-1}) + \mathbf{z_{tr}}$
        $\Delta R = R_t \left[ \beta(\phi_{rot,i} - \phi_{rot,i-1}) + \mathbf{z_{rot}} \right]$
        Update $\mathbf{x}$ with translations $\Delta r$ and rotations $\Delta R$
    **end for**

---

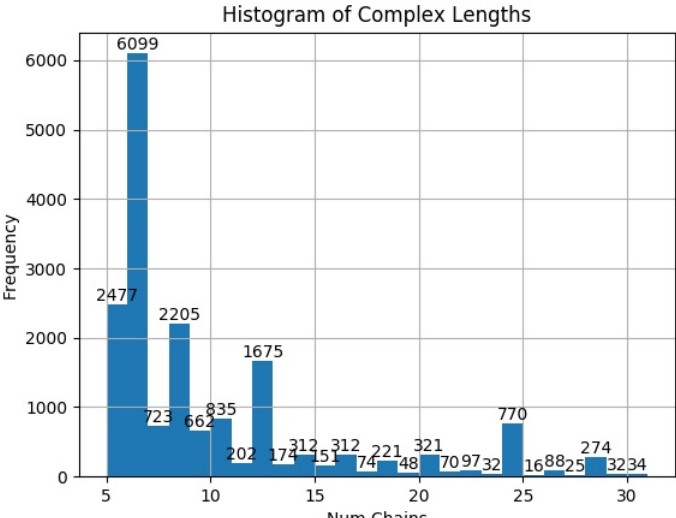

Figure 4: Distribution of complex sizes (in the number of chains) of the trainval dataset.

