# OpenReview forum: "Scalable Multimer Structure Prediction using Diffusion Models"
_NeurIPS.cc/2023/Workshop/AI4Science — NeurIPS2023-AI4Science Poster_

### Official Review · Reviewer_jMpF · 2023-10-08
**Interesting ideas for complex structure modeling**

**Rating:** 5
**Confidence:** 4

**Review:**

1. **Summary**: The authors propose a geometric score-based generative method to rigidly dock individual protein chains into a complex structure formation, with a focus on computational efficiency for scaling up to large complex sizes. The results of the authors' experiments seem to suggest that:
    * (I.) A coarse-grained adaptation of DiffDock for rigid protein-protein docking achieves reasonable performance and excellent completion rates for rigid-body protein docking; and that
    * (II.) Geometric diffusion methods for rigid protein-protein docking (such as those the authors have proposed), in theory, may suffer from high variance in their reverse diffusion process due to the authors' highlighted score degeneracy issue, although, in practice, such methods do not appear to be significantly improved by current solutions to this degeneracy concern.

   Overall, these results are interesting findings for the field and should set the stage for future works to explore even more nuanced questions regarding the behavior of geometric diffusion models in various protein-protein interaction contexts. However, additional comparisons to similar protein-protein docking methods (e.g., in the form of new experiments) are needed.
2. **Strengths and Weaknesses**:
   * Points of strength:
     - The authors' experiments are well-executed and thoughtfully constructed. I especially appreciate the amount of attention the authors paid to the construction of their proposed docking dataset for preventing cross-validation homology leakage.
     - I believe the authors proposed sequence-based coarse-graining is a novel idea that is worth considering in other molecular contexts as well. In other words, the authors' results in this work seem to validate the effectiveness of this kind of coarse-graining procedure for "rigid-body" protein-related tasks.
   * Points for improvement:
     - My main concern is that this work seems (to me) to have a relatively large overlap with DiffDock-PP (https://arxiv.org/abs/2304.03889). In addition to discussing DiffDock-PP directly in the manuscript, I would recommend that the authors consider directly comparing DiffDock-PP to the authors' proposed method to clarify the strengths and weaknesses of each method. I suspect that DiffDock-PP will not scale to large complexes nearly as well as the authors' proposed method, but proper benchmarking is still required to test such a hypothesis.
      - It may also be useful for the authors to consider creating a subset of their validation (or test) dataset with limited "structure-based" similarity to their training dataset, to provide an additional experiment revealing whether or not certain methods only perform well due to "overfitting" to certain protein structural families/folds. One can use tools such as FoldSeek (https://github.com/steineggerlab/foldseek) to accomplish such structure-based filtering.
3. **Recommendation**: Given the authors' promising yet preliminary efforts toward protein complex modeling using recent geometric score-based generative models, I am inclined to **weakly reject** this work.
4. **Rationale behind Recommendation**: Given the limited novelty of the authors' proposed method (i.e., coarse-graining protein chains and then learning a rigid body complex docking diffusion model over the SE(3) group, the latter part being similar to DiffDock-PP in my view), I think a score of 5 for this work is fair.
5. **Questions**:
   (1) Regarding the representation of each chain's structure in a protein complex, the authors mention that they use non-coarse as well as coarse-grained views of each chain. For the non-coarse representation of each protein chain, are they considering all backbone and side-chain atoms in each protein chain in their residue-pooling procedure, or are they instead considering only the Ca atoms for each residue? I believe that clarifying this would be helpful for readers.
6. **Feedback**: I personally find the authors' work innovative from the perspective of scaling up their method to large complexes. It is surprising to me that the proposed method is able to achieve reasonable results after coarse-graining each chain's structure using k-based sequence proximity clustering. This idea may be useful in other contexts (e.g., for different yet related molecular problems).
7. **Submission Type**: The authors' manuscript successfully complies with the four to eight-page requirement for the workshop's submissions. Great work!

---

### Official Review · Reviewer_MHmu · 2023-10-24
**Confusing notations, questions about method details**

**Rating:** 6
**Confidence:** 4

**Review:**

This paper uses diffusion models to generate large multimeric protein structures. The method first predicts the structures of each chain using ESMFold, then generates the rotation and translation of each chain to construct the final complex.

1. The notation makes me confusing. For example, in line 70, there are s chains in the complex, therefore, the complex G should be either {V_0, V_{s-1}} or {V_1, V_s}. In Equation 1, the index n_i in the middle part should be k_i. In line 75 and 76, what is the meaning of the index x?

2. About the method, what is the initial position of each chain? Does the initial position affect the final performance? How can the method deal with the problem of overlap between different chains?

---

### Meta-Review · Area_Chair_kMGa · 2023-10-26

**Recommendation:** Accept (Poster)
**Confidence:** 4

**Metareview:**

The authors propose a score-based method to generate multimeric protein structures. The work has potential to be innovative, however, reviewers have raise concerns, particularly with the notation, lack of clarity of the method, and more thorough benchmark comparisons (eg DiffDock-PP). This work should be of broad interest to the AI4Science community. Recommendation: Poster.